# Fermented Plant Beverages Stabilized with Microemulsion: Confirmation of Probiotic Properties and Antioxidant Activity

Svetlana Merenkova *, Oksana Zinina and Irina Potoroko

Department of Food and Biotechnology, South Ural State University (National Research University),
76 Lenin Avenue, 454080 Chelyabinsk, Russia
* Correspondence: merenkovasp@susu.ru; Tel.: +8-(951)-813-7062

**Abstract:** The purpose of this study was to experimentally confirm the probiotic properties and antioxidant activity of plant fermented beverages stabilized with microemulsion. The object of the study were beverages obtained from hemp seeds and fermented with *Bifidobacterium longum*. To stabilize the plant base, the microemulsion with a bioactive substance (curcumin) was introduced with simultaneous ultrasound treatment. A significant increase in the viscosity of beverages with microcellulose-stabilized microemulsion was noted. Non-fermented plant beverages are characterized by their smaller diameter and distribution of particles in the micro-range, from 0.81 to 6.5 μm. When Twin-stabilized microemulsion was added to beverages, a monodisperse distribution of particles sufficiently small in diameter was observed. A significant increase of 29.4–33.6% in the antioxidant activity of plant beverages stabilized by ME with curcumin was established. A maximum concentration of flavonoids was noted in non-fermented plant beverages containing microemulsion. The results of this study proved the possibility of obtaining fermented plant beverages with identified probiotic and antioxidant properties. A positive effect of stabilizing with a microemulsion loaded with biologically active components on the development of probiotic microorganism cultures in the system of fermented plant products and the formation of their antioxidant activity was established.

**Keywords:** plant beverages; microemulsion; microcrystalline cellulose; ultrasound treatment; probiotic microorganisms; functional properties; antioxidant activity

## 1. Introduction

Traditional widely consumed dairy products are distinguished by a balanced nutrient composition and a unique list of items, in accordance with the consumption culture of different countries. However, in certain groups of the population, the consumption of products containing cow's milk components can cause undesirable allergic reactions, which can trigger the occurrence of serious pathologies [1]. Lactose intolerance has a global prevalence; it affects approximately 60% of the world's population. This disease is most common in South America, Africa, and Asia. In Europe, the prevalence of lactose malabsorption is around 28% [2,3]. These factors determine the necessity for developing a range of new types of products, including beverages made from plant raw materials, including cereals, oilseeds, legumes, or nuts.

Numerous scientific publications have presented high-tech methods for producing beverages based on plant raw materials [4,5]. For example, the use of a microwave for the production of milk substitutes based on nut kernels is experimentally justified [6]. The effectiveness of ultrasonic (US) processing for obtaining protein drinks based on semi-fat flour from the kernel of pine nuts has been proven [7]. The principal technological scheme for obtaining oat milk also has been proposed [8]. It has been established that the use of US enables emulsions to be made with different dry matter and soluble protein contents [7].

Scientific publications have provided data on the health-protective properties of probiotic products due to their positive effects on the digestive tract, cardiovascular system, and immune system [9,10].

The biological activity of grain processing products can be increased by applying biotechnological approaches. Microbial fermentation of the plant base enables the bioavailability and digestibility of essential components to be enhanced. The selection of industrial microorganisms enables products with probiotic properties to be produced [11–13]. Fermented plant products are characterized by a significant content of bioactive, essential nutrients; improved organoleptic properties; and a decrease in the concentration of toxins and anti-nutritional compounds [14–16].

The rational selection of mono- and multi-strain starter cultures of microorganisms can help improve probiotic properties. Many researchers have determined and substantiated that fermented beverages from plant raw materials have optimal quality indicators, are characterized by probiotic properties, and can be recommended for people with lactose intolerance [17,18].

The main function of probiotic microorganisms is to restore the balance of the normal microflora of the gastrointestinal tract. An imbalance in the intestinal microbiota causes changes in the production of neuronal molecules and leads to neurodegenerative diseases [19]. Probiotic products contribute to the restoration of intestinal eubiosis, improve its functions, stimulate immune responses, and regulate the activity of the nervous system [20].

There are several limitations in the industrial-scale production of plant-based drinks with high nutrient adequacy. Thus, when using high-oil raw materials, the problem of maintaining the emulsion stability and resistance to oxidative deterioration appears. In industrial production, food additives are used to solve the problem. Vegetable oils are additionally added, and various plant bases are combined to obtain stable compositions. Oilseed proteins can exhibit emulsifying and stabilizing properties, which are enhanced by US exposure. Oilseed extracts are a favorable prebiotic basis for the development of lactic acid microorganisms [16,21].

When producing plant beverages, an important aspect is the effective extraction of biologically active components from raw materials and the preservation the compounds stability during the recommended shelf life. The colloidal and emulsion stability of food systems is a result of several factors (Figure 1).

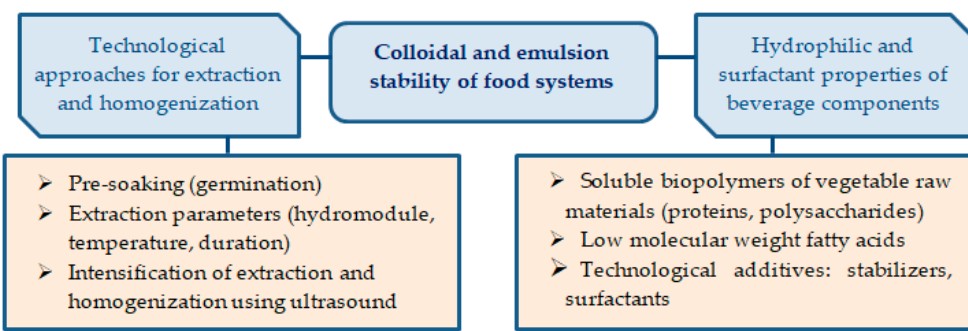

**Figure 1.** Factors providing colloidal and emulsion stability in plant beverages.

To stabilize the nutritional value of plant-based drinks, biologically active components with antioxidant activity can be added into the composition. This increases the resistance of the product to oxidative processes. The introduction of bioactive substances in the composition of microemulsions (ME) into the food system of the beverage can increase the stability and bioavailability of minor compounds [22].

The use of ME loaded with bioactive components and stabilized with a biopolymer into the composition of plant beverages suggests a wide range of positive effects, including an increase in colloidal stability and inhibition of radical-oxidative reactions [23]. The stabilization abilities of ME by creating a physical barrier through solid low-molecular colloidal particles is promising. It reduces interfacial tension and enables functional Pickering emulsions to be obtained. To increase the system stability in the Pickering emulsion technology, the use of low-frequency US treatment is recommended [24].

The use of natural macromolecular compounds, in particular microcrystalline cellulose (MCC), as stabilizers of the microemulsions is promising [25,26]. The mechanism of aggregative stabilization of the microemulsion, and thus plant drinks, depends on the surface-active properties of MCC (adsorption on the surface of lipid droplets and inhibition of their coalescence) and hydrophilic properties (increase in the viscosity of the medium [27].

Thus, plant beverages fermented by industrial cultures of microorganisms and stabilized by microemulsion, can be recommended as specialized nutrition for persons with lactose and casein intolerance, as well as a functional product—a source of probiotics.

The purpose of this study was to experimentally confirm the probiotic properties and antioxidant activity of plant fermented beverages stabilized with microemulsion.

## 2. Materials and Methods

### 2.1. Raw Materials and Ingredients

Hemp seeds of the Nadezhda variety, 2021 harvest, were used in the production of plant beverages. The approximate chemical composition of the seeds is as follows: protein content 22.1%; fat 30.6%; carbohydrates 16.9%; fiber content 21.8%. The nutritional composition of hemp seeds produced by the enterprise "Medal" was analyzed in the testing center of the Federal State Educational Institution "KubGTU" (Russia). Seeds were stored before the start of research at a temperature of 20 °C for no more than 3 months.

To obtain a stabilizing microemulsion (ME) (Pickering emulsion), a number of components were used: hemp oil obtained by cold pressing from hemp seeds; curcumin extract in powder form containing 10% of the active substance (Rajapuri turmeric finger: India.); emulsifier Twin-80 (Italy); and microcrystalline cellulose (MCC)—P-2019/USP-41 with a particle size of 200 mesh (70–80 microns) (Silverline chemicals, India). Supplements were purchased from a distributor 100ING.RU.

Bacterial liquid concentrate of *Bifidobacterium longum* B 379M with activity $10^{11}-10^{12}$ CFU/cm$^3$ (Propionics LLC, Russia) was used for microbiological fermentation.

### 2.2. Manufacturing of Fermented Plant Beverages

The technology for obtaining a plant base consisted of soaking the seeds for 24 h, grinding them, coarse filtration, homogenization at 1800 rpm for 5 min (Stegler DG-360), repeated fine filtration on sieves with a mesh size of up to 100 microns, followed by pasteurization at 70 °C (as described in previous studies) [28].

Microemulsions loaded with BAS (curcumin extract) were prepared. To obtain a loaded O/W type microemulsion, curcumin was mixed with hemp oil for 4 h at 100 rpm. Microcellulose or Tween was added to the water as a stabilizer. US treatment was carried out, then the oil phase was gradually added with constant US homogenization. As a working tool, an ultrasonic apparatus "VOLNA-L" UZTA-0.63/22-OL, (Russia) with a working tool of a submersible type was used. The treatment was carried out under the following US regimes: power 630 W, frequency $(20 \pm 2)$ kHz for 20 min, with temperature control not exceeding 40 °C. The ratio of components of microemulsions O/W was as follows: oil 10%; BAS 2%; stabilizer 5%.

To stabilize the plant base, the resulting ME with biologically active substances was introduced in an amount of 5% by weight of the beverage with simultaneous pulsed US treatment (frequency $(12 \pm 2)$ kHz; duration of 1 min—twice).

Bacterial liquid concentrate of *Bifidobacterium longum* was used to ferment ME-stabilized beverages. The thermophilic strains of microorganisms were introduced in the amount of 3% (which corresponds to $1 \times 10^6$ CFU/g). The beverages were fermented at 38–40 °C for 10–12 h until a weak clot was formed and a pH level below 4.7 was achieved. The beverages were cooled to $(4 \pm 2)$ °C and stored for 7 days (Figure 2).

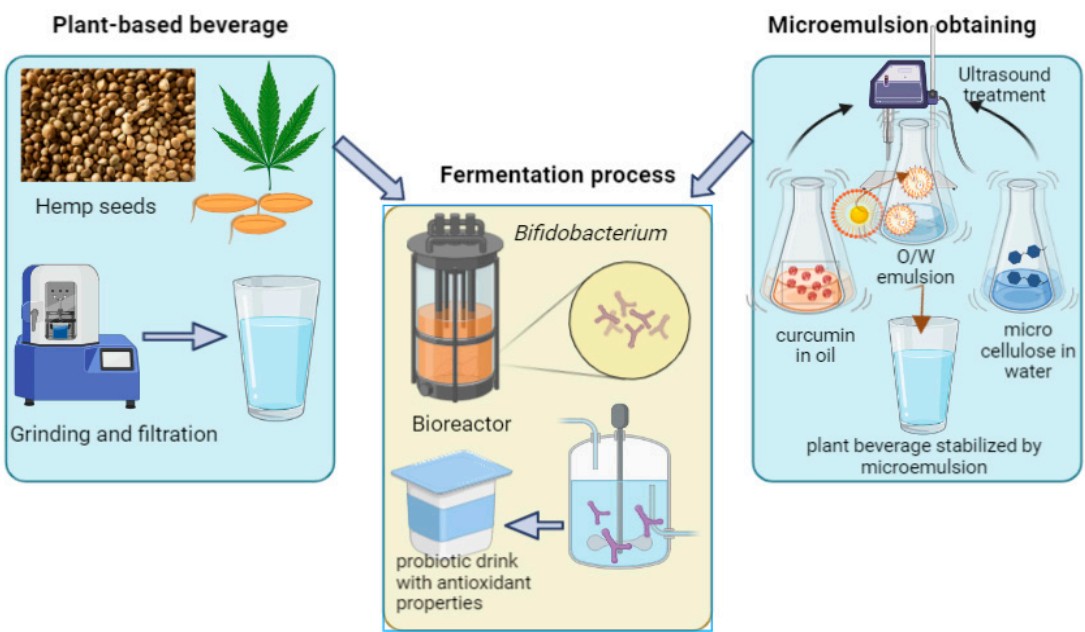

**Figure 2.** Technological stages for obtaining fermented drinks stabilized with ME. (Created with BioRender.com).

The ingredients in the samples of plant beverage formulation is presented in Table 1.

**Table 1.** Ingredients of the plant beverages formulation.

| Designation of Samples | Content of the Plant Base from Hemp Seeds, % | Content of the ME with Curcumin Stabilized by MCC % | Content of the ME with Curcumin Stabilized by Twin-80 % | Content of *Bifidobacterium longum* Concentrate, % |
|---|---|---|---|---|
| Bev Contr | 100 | – | – | – |
| Bev Bif | 100 | – | – | 3 |
| Bev MCC | 95 | 5 | – | – |
| Bev Twin | 95 | – | 5 | – |
| Bev Bif+ MCC | 95 | 5 | – | 3 |
| Bev Bif +Twin | 95 | – | 5 | 3 |

Designation of samples: Bev Contr—unfermented beverage from hemp seeds, Bev Bif—beverage from hemp seeds, fermented by *Bifidobacterium longum*; Bev MCC (Bev Twin)—unfermented beverage from hemp seeds, stabilized by microemulsion with microcellulose (twin-80); Bev Bif+ MCC (Bev Bif +Twin)—beverage from hemp seeds, stabilized by microemulsion with microcellulose (twin-80) fermented by *Bifidobacterium longum*.

*2.3. Methods of Analysis*

2.3.1. Analysis of Viscosity and Biochemical Composition

We determined the intensity of metabolic processes in the samples obtained during fermentation: active acidity (pH level), dynamic viscosity, content of lactic acid, dry matter, and protein concentration.

The active acidity of the samples was determined by immersing the electrode of a multiparameter stationary pH meter (edge HI 2002-02 by Hanna Instruments, Romania) in the beverage emulsion for 2 min. The pH level was studied every 2 h during fermentation. Lactic acid accumulation was determined by titrimetric analysis of 0.1 N NaOH using phenolphthalein (0.1% wt./vol. in 95% ethanol) as an indicator, which was subsequently converted to lactic acid concentration (g/100 mL).

The dynamic viscosity of the samples was determined using an AND SV vibro viscometer. The measurement was carried out for 60.0 s at $(22.0 \pm 2.0)$ °C.

The content of dry matter was measured using a digital refractometer Abbemat 500 (Anton Paar, Austria). Total nitrogen content was assayed by the Kjeldahl method with

nitrogen converted to equivalent protein content using a factor of 6.25 (Methods 992.15 and 992.23).

### 2.3.2. Investigation of Flavonoid Content and Antioxidant Activity

Extraction of Phenolics

Beverage samples were extracted with 5-fold 80% (*v/v*) ethanol in a water bath at 40 °C for 3 h. The solution extracted was centrifuged at $8000 \times g$ for 15 min, and the supernatant was evaporated to dryness using a rotary evaporator at 50 °C [29]. The phenolic extracts were then redissolved in 80% (*v/v*) ethanol for further analysis (phenolic samples).

Antioxidant Activity Analysis

DPPH radical (DPPH·) radical scavenging activity was determined according to the method of Xiao, Rui, et al. [29]. Specifically, 2 mL of phenolic sample was added to 2 mL of DPPH solution (0.4 mmol/L), and the mixture was allowed to stand in the dark for 30 min. The absorbance was then recorded at 515 nm.

Determination of Total Flavonoid Content (TFC)

The contents were spectrophotometrically measured based on the formation of a flavonoid–aluminum complex. One milliliter of sample was mixed with 0.10 mL of 5.0% $NaNO_2$ for 6.0 min. Then, 0.10 mL of 10.00% $AlCl_3 \cdot 6H_2O$ solution was added to the mixture for another 5.0 min. After adding 1.0 mL of 1.0 mol/L NaOH, the reaction solution was mixed well and allowed to stand for 15 min. The absorbance was measured at 510 nm. Quercetin was used as a standard to establish the calibration curve. The TFCs were calculated and expressed in quercetin equivalent, i.e., mg EQ/g dry weight DW.

### 2.3.3. Determination of Dispersed Composition

The study of the dispersed composition and the analysis of the particle size in the samples were carried out using the method of laser dynamic light scattering on a laser diffraction analyzer Microtrac S3500 (Microtrac MRB, Osaka, Japan). Program: Microtrac FLEX 10.6.1.

### 2.3.4. Analysis of Bioavailability and Membrane Stabilizing Properties

The digestibility criteria and membrane stabilizing properties of drinks were determined by the biotesting method using infusoria of the species *Parametium caudatum* according to the method proposed by Stepanova et al. [30]. The counting of infusoria cells was carried out using the "BioLaT-3.2" device (Pushkino, Moscow region, Russia) in the short-term mode.

### 2.3.5. Analysis of Probiotic Properties and Microbiological Investigations

GMK-1 corn-lactose medium (NPC Biokompas-S, Uglich, Russia) was used to determine the amount of probiotic microorganisms. Before the study, the medium was heated in a boiling water bath until the agar was completely melted. At the time of use, the temperature of the nutrient medium was $38 \pm 1$ °C. One milliliter of each beverage sample was added to 9.0 mL of 0.85% sterile saline (sodium chloride) and further dilutions were made. From the last four dilutions of the product, inoculations of 1 mL were made in two parallel rows of test tubes with a nutrient medium. After incubation at 37 °C for 72 h, the number of colonies was counted. The last tubes in which colonies had grown were used for counting. Confirmation of the presence of *Bifidobacterium* was carried out by microscopy (Micromed-1, Saint Petersburg, Russia). For this purpose, smears were prepared from isolated colonies and stained by Gram. The number of viable cells was expressed in CFU/g of samples. The count of microorganisms was carried out in triplicate. Microbiological investigations were carried out after the end of the fermentation of the beverages and after 7 days of storage.

To assess the microbiological safety, the content of conditionally pathogenic microorganisms, yeast, and mold was calculated in the beverage samples after fermentation as well as after 7 days of storage. Enumeration of total coliforms, Salmonella sp., mold, and yeasts was performed using Petritest™ microbiological rapid tests (Research and Production Association "Alternativa", Russia).

A 10 g drink sample was taken to prepare the initial dilutions. Salt solution was used for dilution and 0.2 cm$^3$ of the dilution was applied to the open surface of the test substrate. The lid was closed with latches, and the test liquid was evenly distributed over the surface of the nutrient medium. Inoculated plates were incubated at 30◦C for 24 h for bacterial counts. To count the number of yeasts and molds, the plates were incubated at 25 °C for 24 h (for preliminary counting) and 120 h (for final counting). The result was multiplied by the value of the appropriate dilution and the total number of viable bacteria or fungi per 0.2 cm$^3$ of the sample was obtained. The results were multiplied by five to bring the results to 1 cm$^3$ according to the manufacturer's recommendations.

### 2.4. Statistical Analysis

Analyses were performed on five replicates. The results were expressed as the mean values of the five replicates $\pm$ the standard deviation. Probability values of $p \leq 0.05$ were taken to indicate statistical significance. The data was analyzed via one-way ANOVA analysis of variance using the free web-based software offered by Assaad et al. [31].

## 3. Results and Discussion

### 3.1. Analysis of Viscosity and Biochemical Parameters of Plant Beverages Stabilized with ME

The dynamic viscosity of beverages is an indirect indicator that characterizes the level of metabolic activity of microorganisms during fermentation. Viscosity was detected on the first day, immediately after fermentation (preparation) of beverages, and within 7 days of storage. For non-fermented samples stabilized by ME with Twin, a slight decrease in viscosity was found when compared to that of the beverage without microemulsion (Bev Contr), to a level of 1.62–1.92 mPa·s. During the fermentation of beverages containing ME with Twin-80, the viscosity values were comparable to the viscosity of fermented beverage without ME (Bev Bif). This remained stable throughout the entire storage period.

When microemulsion stabilized with microcellulose was introduced into the beverage, a significant increase in the viscosity of both fermented (Bev Bif + MCC) and unfermented (Bev MCC) samples to 2.27–3.46 mPa·s was noted on the first day. A further increase in this indicator to 5.29–5.76 mPa·s was then found. This trend is associated with the hydrophilic properties of microcellulose, as well as the ability of microorganisms to accumulate exopolysaccharides during the lag phase. Maximum viscosity values for Bev Bif + MCC sample were established, which were associated with the ability of *Bifidobacterium* to efficiently produce exopolysaccharides (EPS) in a food system containing prebiotic components (Figure 3).

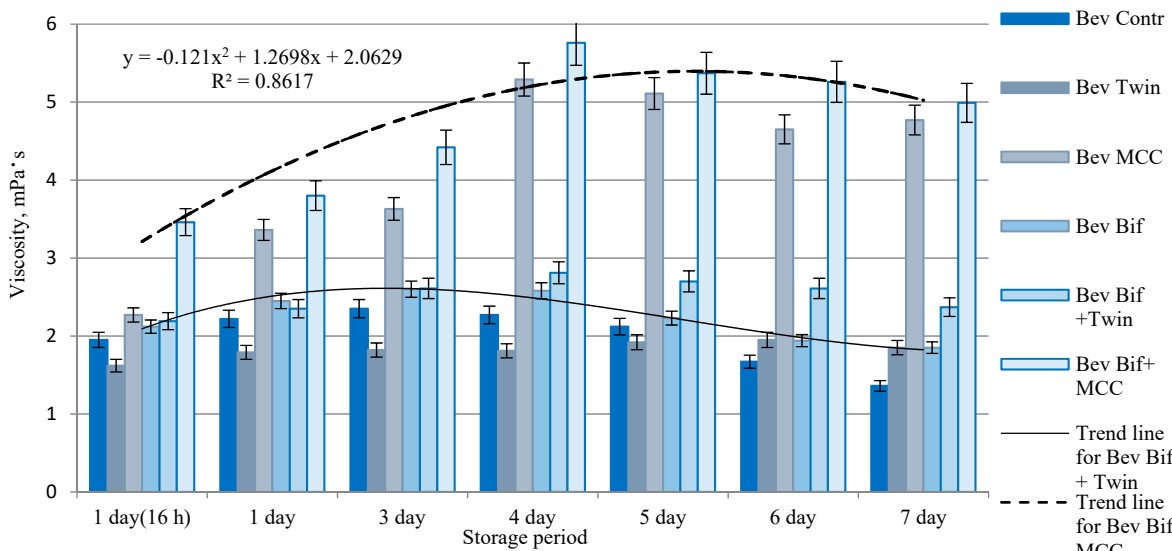

**Figure 3.** Dynamic viscosity of plant beverages stabilized with ME. The error bars represent the standard deviation of measurements (*n* = 5). Designation of samples: Bev Contr—unfermented beverage from hemp seeds, Bev Bif—beverage from hemp seeds, fermented by *Bifidobacterium longum*; Bev MCC (Bev Twin)—unfermented beverage from hemp seeds, stabilized by microemulsion with microcellulose (twin-80); Bev Bif+ MCC (Bev Bif +Twin)—beverage from hemp seeds, stabilized by microemulsion with microcellulose (twin-80) fermented by *Bifidobacterium longum*.

The introduction of the microemulsion had a significant effect on the biochemical composition of the beverages, and changed the enzymatic activity of microorganisms. During the fermentation of beverages, an active accumulation of lactic acid was found. This caused a shift in the pH level to the acidic side (3.75–4.65) for samples of fermented beverages. The stability of this indicator was noted during the storage period of 7 days. Due to the presence of these bacterial metabolic products, the shelf life increases due to a decrease in pH [32,33].

During the process of metabolism, *Bifidobacterium* effectively produce various exo-components into the nutrient medium (protein, EPS), and as a result, the content of dry substances in the beverages increases. Moreover, the efficiency of the accumulation of components is associated with the prebiotic properties of stabilizers including in ME and BAS. The most significant increase in the concentration of dry substances and protein was observed in this case of beverages fermented with *Bifidobacterium* containing ME with microcellulose and ME with Twin-80 (Table 2).

**Table 2.** Biochemical parameters of plant beverages stabilized with ME.

| Designation of Samples | Indicators | | | |
|---|---|---|---|---|
| | Dry Matter Content, % | Protein Content, % | pH Level | Lactic Acid Content, g/100 mL |
| Bev Contr | 5.98 ± 0.12 [a] | 2.05 ± 0.09 [a] | 6.70 ± 0.12 [c] | 1.21 ± 0.05 [a] |
| Bev Bif | 6.48 ± 0.14 [b] | 2.35 ± 0.11 [b] | 3.75 ± 0.08 [a] | 5.22 ± 0.12 [c] |
| Bev MCC | 7.41 ± 0.18 [c] | 2.27 ± 0.10 [b] | 6.90 ± 0.12 [c] | 1.31 ± 0.07 [a] |
| BevTwin | 7.80 ± 0.16 [cd] | 2.30 ± 0.08 [b] | 6.65 ± 0.10 [c] | 1.40 ± 0.05 [a] |
| Bev Bif+ MCC | 8.21 ± 0.20 [d] | 2.86 ± 0.09 [c] | 4.40 ± 0.09 [b] | 3.51 ± 0.08 [b] |
| Bev Bif +Twin | 7.72 ± 0.14 [cd] | 2.92 ± 0.15 [c] | 4.55 ± 0.08 [b] | 4.16 ± 0.14 [b] |

The values are means ± standard deviation of five replicates. Different letters in the same column refer to a significant difference at (*p* ≤ 0.05). Designation of samples: Bev Contr—unfermented beverage from hemp seeds, Bev Bif—beverage from hemp seeds, fermented by *Bifidobacterium longum*; Bev MCC (Bev Twin)—unfermented beverage from hemp seeds, stabilized by microemulsion with microcellulose (twin-80); Bev Bif+ MCC (Bev Bif +Twin)—beverage from hemp seeds, stabilized by microemulsion with microcellulose (twin-80) fermented by *Bifidobacterium longum*.

*3.2. Dispersed Composition of the Plant Beverages Stabilized with ME*

Plant-based beverages are complex food systems with fine particles dispersed in an aqueous environment in the form of fat droplets, protein particles, and fragments of plant cells [34]. The structure and size of these colloidal particles ultimately determines the technological, functional, organoleptic, and nutritional properties of plant beverages, including the bioavailability of individual nutrients.

Non-fermented plant beverages (Bev Contr, Bev Twin, Bev MCC) are characterized by a smaller diameter and distribution of particles in the micro-range, from 0.81 to 6.5 μm (75–90% of particles). For these samples, the smallest average hydrodynamic particle diameter was noted in the range of 3.36–7.93 μm. During fermentation, protein particles adhere in connection with the formation of a bound matrix of a probiotic product. This relationship is stabilized by exopolysaccharides of microorganisms and the introduced emulsion with microcellulose. For beverages fermented by *Bifidobacterium* without emulsion, 70–85% of the particles were distributed in size from 7.7 to 37 microns. When a microcellulose-stabilized microemulsion was introduced, the particles distribution shifts to the micro-range from 1.9 to 26 μm (87% of particles). When Twin-stabilized microemulsion was added to the beverages, a monodisperse distribution of particles sufficiently small in diameter was observed for Bev Twin and Bev Bif Twin samples. The concentration of particles with a diameter from 0.81 to 1.63 microns averaged 44–53%. The maximum hydrodynamic particle diameter was established for fermented beverage particles not containing microemulsion (Bev Bif 19.13μm) (Figure 4).

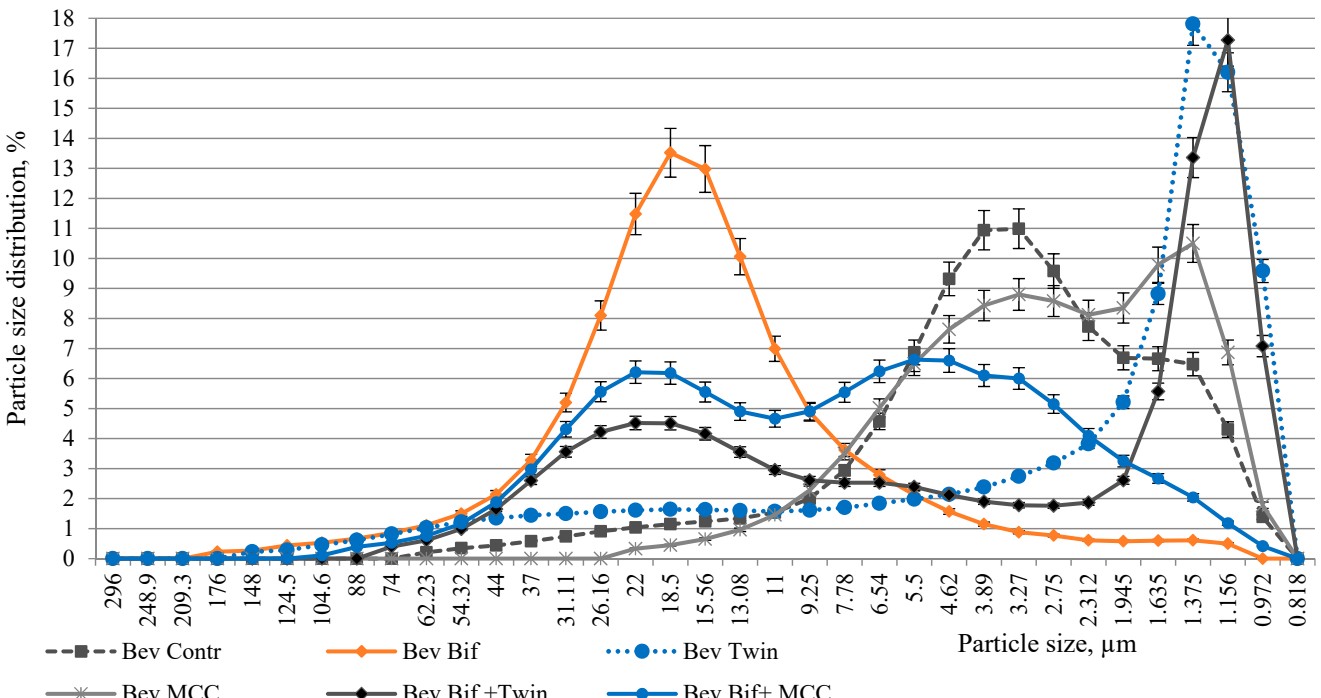

**Figure 4.** Particle size distribution for microemulsion-stabilized plant beverages. The error bars represent the standard deviation of measurements (*n* = 5). Designation of samples: Bev Contr—unfermented beverage from hemp seeds, Bev Bif—beverage from hemp seeds, fermented by *Bifidobacterium longum*; Bev MCC (Bev Twin)—unfermented beverage from hemp seeds, stabilized by microemulsion with microcellulose (twin-80); Bev Bif+ MCC (Bev Bif +Twin)—beverage from hemp seeds, stabilized by microemulsion with microcellulose (twin-80) fermented by *Bifidobacterium longum*.

Numerous scientific studies have established the characteristics of the particles contained in plant beverages. Their stability depends on the technological processes and parameters used in their production technology [34,35]. For example, to reduce the particle

size and unify the structure, plant-based beverages are homogenized [36]. US is also one of the most popular methods of preparing nano- and microemulsions [37,38]. The physical forces created by ultrasonic cavitation can enhance the destruction of oil droplets, thereby contributing to the formation of a stable microemulsion with monodisperse droplets with a diameter of less than 100 nm [38]. The size of the emulsion droplets affects digestion, absorption, and the bioavailability of the components. The average droplet surface diameter is especially important, since a lower value of this indicator provides a larger surface area available for digestive enzymes [39].

*3.3. Antioxidant Activity and Flavonoid Content in Plant Beverages Stabilized with ME*

The nutritional benefit of fermented plant beverages filled with dispersed bioactive components, which are bioavailable for starter microflora, in particular for *Bifidobacterium*, has been proven. A significant part of the components of the initial raw materials and those accumulated in drinks during the fermentation process is known to exhibit adaptogenic properties, including antioxidants. Curcumin and microcellulose used in beverage technology provide pronounced antioxidant properties. The study of the antioxidant activity of the samples was carried out on the ability to capture free stable radicals by the radical DPPH (1,1-diphenyl-2-picrylhydrazyl). The method is widely used to evaluate the antioxidant activity of individual phenolic substances and food systems in general. The DPPH test is more selective than the currently used ABTS test (TEAC protocol with the radical 2,2′-azinobis (3-ethylbenzothiazoline-6-sulfonate)) [40].

While ensuring the availability of components (microstructured by US treatment curcumin and cellulose) in the composition of ME, it is necessary to establish the risks of reducing antioxidant activity and determine the most rational approaches to preserve benefit properties. A significant increase in the antioxidant activity of plant beverages by 29.4–33.6% ($p \leq 0.05$) was established when a stabilizing ME with curcumin was added compared with the control sample.

When we consider the initial properties of the beverage components, when cellulose exhibits hydrophilic properties, and curcumin exhibits hydrophobic properties (Figure 2), under controlled non-thermal exposure modes of US, we obtain an emulsion that allows us to reveal the antioxidant potential of phytocomponents in the composition of the microemulsion.

In addition, the fermentation of beverages with ME improved their antioxidant properties by 1.0–2.8% ($p \leq 0.05$) when compared to non-fermented drinks with the same composition. The maximum content of flavonoids was established in non-fermented plant beverages containing ME.

After 7 days of storage, a decrease in the concentration of flavonoids b 32.9–35.5% was observed, as well as a reduction in antioxidant activity by 12.1–12.3% for fermented beverages stabilized with ME (Table 3).

Results are confirmed in studies by other authors who establish the considerable antioxidant activity of non-dairy probiotic drinks [41]. Rocchetti et al. confirmed that microbial fermentation is effective in increasing flavonoid content and antioxidant potential in cooked quinoa seeds. They also showed that fermentation induces the release of certain classes of phenols [42]. Wu et al. proved that oats fermented with *Lactobacillus plantarum* and *Rhizopus oryzae* had higher total phenol content than unfermented oats, and its DPPH radical scavenging activity was significantly increased [43].

**Table 3.** Antioxidant activity and flavonoid content in plant-based beverages stabilized with ME.

| Designation of Samples | Indicators | | | |
|---|---|---|---|---|
| | Content of Flavonoids, μg EQ/g (1st Day) | Content of Flavonoids, μg EQ/g (7th Day) | DPPH Activity, % (1st Day) | DPPH Activity, % (7th Day) |
| Bev Contr | 9.2 ± 0.8 [a] | 6.5 ± 0.4 [a] | 47.63 ± 0.60 [a] | 34.81 ± 0.55 [a] |
| Bev Bif | 10.9 ± 0.9 [a] | 8.1 ± 0.5 [ab] | 61.65 ± 1.15 [b] | 52.70 ± 0.70 [b] |
| Bev MCC | 33.0 ± 1.1 [d] | 16.6 ± 0.8 [c] | 63.72 ± 1.35 [c] | 55.13 ± 0.72 [c] |
| Bev Twin | 35.1 ± 1.0 [d] | 22.8 ± 1.1 [d] | 64.82 ± 0.85 [cd] | 55.36 ± 0.65 [cd] |
| Bev Bif+ MCC | 17.6 ± 0.7 [b] | 11.8 ± 0.8 [b] | 65.53 ± 1.50 [d] | 56.91 ± 0.67 [d] |
| Bev Bif +Twin | 21.4 ± 0.9 [c] | 13.8 ± 1.0 [c] | 65.41 ± 1.10 [d] | 56.74 ± 0.85 [d] |

The values are means ± standard deviation of five replicates. Different letters in the same column refer to a significant difference at ($p \leq 0.05$). Designation of samples: Bev Contr—unfermented beverage from hemp seeds, Bev Bif—beverage from hemp seeds, fermented by *Bifidobacterium longum*; Bev MCC (Bev Twin)—unfermented beverage from hemp seeds, stabilized by microemulsion with microcellulose (twin-80); Bev Bif+ MCC (Bev Bif +Twin)—beverage from hemp seeds, stabilized by microemulsion with microcellulose (twin-80) fermented by *Bifidobacterium longum*.

### 3.4. Analysis of Bioavailability and Membrane Stabilizing Properties

The method proposed by Stepanova et al. was used [30] for the indirect analysis of the bioavailability of bioactive components in the composition of plant beverages. Survival rates and relative growth of infusoria were determined using cultures of protozoa *Paramecium caudatum*. The counting of infusoria cells was carried out using a "BioLaT-3.2" device in the short-term mode. As a result of the bioavailability analysis, it was found that the number of viable infusoria after 3 h of exposure increased for the control beverages by 113–125.8%, and for beverages stabilized by ME by 150–271%, depending on the implementation of the fermentation stage. The maximum values of the digestibility criterion are achieved for fermented plant beverages stabilized by ME. The maximum bioactivity of fermented beverages with microemulsion was achieved by both the bioactive properties of phytocomponents of initial materials and antioxidants (curcumin) introduced with microemulsion. In addition, because of fermentation, an accumulation of adaptogenic components is observed, which are the metabolism products of probiotic bacteria (Figure 5).

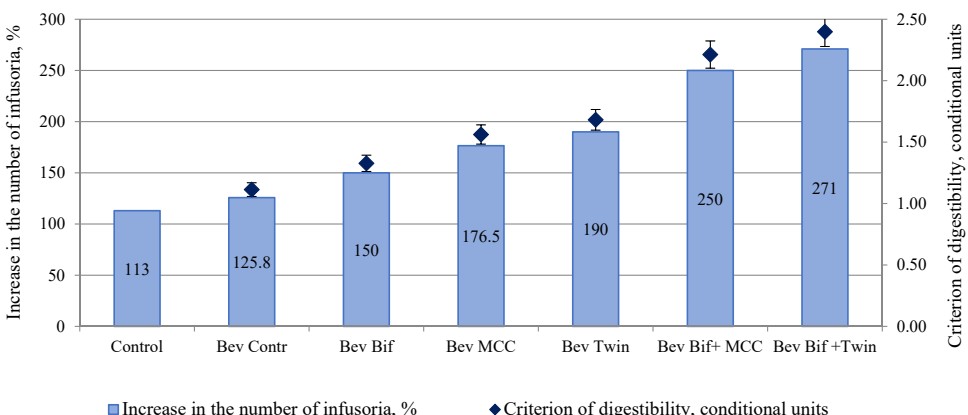

**Figure 5.** Criterion of digestibility of plant beverage samples in biotesting for *Parametium caudatum*. The error bars represent the standard deviation of measurements ($n$ = 5). Designation of samples: Control—the well with a nutrient medium; Bev Contr—unfermented beverage from hemp seeds, Bev Bif—beverage from hemp seeds, fermented by *Bifidobacterium longum*; Bev MCC (Bev Twin)—unfermented beverage from hemp seeds, stabilized by microemulsion with microcellulose (twin-80); Bev Bif+ MCC (Bev Bif +Twin)—beverage from hemp seeds, stabilized by microemulsion with microcellulose (twin-80) fermented by *Bifidobacterium longum*.

To indirectly determine the membrane-stabilizing effect of plant beverages, an express method using infusoria was used. *Paramecium caudatum* have cilia on their entire surface that act as chemoreceptors that react to dissolved chemicals. Indirectly, adaptogenic activity can be judge by the ability to increase the tolerance of paramecia to cellular poisons under the influence of the studied substances. Infusoria are in constant motion, so it is easy to observe the slightest changes in movement under the influence of poison [44,45].

A 3% hydrogen peroxide solution was used as a toxicant in the study of membrane stabilizing activity. An equivalent volume of the plant beverage was added to the paramecium culture for 60 min (the period when protective mechanisms are formed), then a pathological model of paramecium membrane damage with a 3% hydrogen peroxide solution was created. Their number was counted in dynamics for 60 min. Intact paramecium cells served as controls (Figure 6).

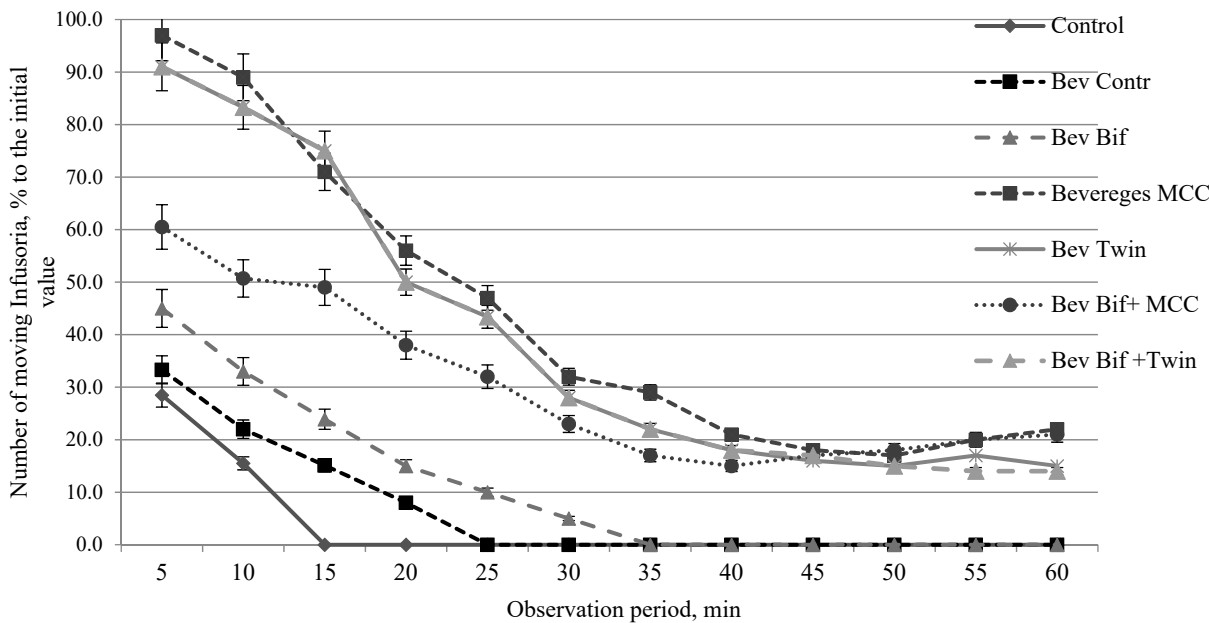

**Figure 6.** Membrane-stabilizing properties of plant beverages stabilized with ME in biotesting for *Parametium caudatum*. The error bars represent the standard deviation of measurements (*n* = 5). Designation of samples: Control—the well with a nutrient medium; Bev Contr—unfermented beverage from hemp seeds, Bev Bif—beverage from hemp seeds, fermented by *Bifidobacterium longum*; Bev MCC (Bev Twin)—unfermented beverage from hemp seeds, stabilized by microemulsion with microcellulose (twin-80); Bev Bif+ MCC (Bev Bif +Twin)—beverage from hemp seeds, stabilized by microemulsion with microcellulose (twin-80) fermented by *Bifidobacterium longum*.

Analysis of the data obtained allows us to conclude that fermented plant beverages stabilized by ME with microcellulose and Tween, as well as non-fermented drinks containing ME, have a pronounced membrane-stabilizing effect. For these beverage samples, the survival rate of infusoria cells was observed to be 14–22% after 60 min following the introduction of the toxicant. Death of all infusoria cells for control plate occurred in 15 min after the injection the poison, and the death of protozoa for the control beverages without the addition of a microemulsion occurred after 25 min.

### 3.5. Analysis of Probiotic Properties and Microbiological Investigations of Plant Beverages Stabilized with ME

Active development of beneficial microflora occurs in the process of fermentation. This leads to an improvement in the nutritional value and organoleptic properties due to the accumulation of metabolites such as organic acids, vitamins, bacteriocins, and biologically active substances. Probiotics support the synthesis of vitamins and promote the absorption

of calcium [32]. Peiroten et al. noted the potential of lactic acid bacteria and *bifidobacteria* in the production of a fermented soy drink enriched with bioactive isoflavones [46].

As a result of fermentation, the number of viable *Bifidobacterium* in all beverage samples increased compared to the inoculated amount, and slightly decreased after 7 days of storage (Table 1). However, they remained at the level required for probiotic products—at least $1 \times 10^7$ CFU/mL [47]. Fermented beverages prepared with probiotic cultures can be considered as probiotic products only if the microbes are still live cells at the time of consumption [48]. Langa et al., in a study of fermented soy drinks with *B. pseudocatenulatum INIA P815* (B.815), showed that the initial strain concentration was maintained or doubled after 3 days of storage and was at the level of $10 \times 10^7$ to $20 \times 10^7$ CFU/mL [49]. It was found that the largest number of viable populations after fermentation of rice-based beverages was obtained for *Lactobacillus* sp. ($3 \times 10^7$–$1.7 \times 10^8$ CFU/mL) and *B. animalis* subsp. *lactis* ($1.9$–$3.2 \times 10^8$ CFU/mL). In most of the tested samples, a significant decrease in the viable population was observed already after 14 days of storage (Table 4) [47].

**Table 4.** Probiotic microorganism content in samples of plant fermented beverages.

| Indicator | Beverage Sample/Probiotic Microorganism Content, Log CFU/mL | | |
|---|---|---|---|
| | Fermented Beverage without Emulsion | Fermented Beverage with MCC Stabilized Emulsion | Fermented Beverage with Tween stabilized Emulsion |
| After fermentation | $7.42 \pm 0.02$ [b] | $8.08 \pm 0.03$ [a] | $7.91 \pm 0.03$ [c] |
| After 7 days of storage | $7.08 \pm 0.02$ [a] | $7.49 \pm 0.02$ [b] | $7.20 \pm 0.03$ [a] |

The values are means ± standard deviation of five replicates. Different letters in the same row refer to a significant difference at ($p \leq 0.05$).

Pathogenic microorganisms that may be present in foods (such as coliforms, *Salmonella* sp., molds, etc.) not only make them dangerous, but also affect the nutritional value. They reduce the content of active components and worsen the organoleptic properties [50]. After the fermentation of beverages, as well as after 7 days of storage, all samples were free of bacteria of the *Escherichia coli* group, *Salmonella* sp., yeast, and mold, indicating the microbiological safety of the products studied. There is evidence that probiotic bacteria such as *Lb.*, *Leuconostoc*, *Pediococcus*, and *Bifidobacterium* have a damaging effect on pathogenic genera such as *Clostridium*, *Salmonella*, *Shigella*, *Escherichia*, *Helicobacter*, *Campylobacter*, *Candida*, etc. [51].

## 4. Conclusions

The results of this study prove the possibility of obtaining fermented plant beverages with identified probiotic and antioxidant properties. The data obtained showed the positive effect of a stabilizing microemulsion loaded with biologically active components in the development of probiotic microorganism cultures in the system of fermented plant products and the formation of their antioxidant activity. The stabilization of emulsions with microcellulose additionally loaded with curcumin due to their prebiotic properties stimulated the development of beneficial microflora in beverages. In the study, this was experimentally proved by analyzing the viability of probiotic microorganisms (*Bifidobacterium*).

To create sustainable stabilized microemulsions on the principle of Pickering emulsions, the dispergation method of the beverage system can be used at particular stages of production based on non-thermal ultrasound exposure with the application of stabilizers (plant hydrocolloids). The resulting microemulsion adequately exhibited the properties of surfactants in the composition of fermented drinks. Furthermore, it has prebiotic properties, and reduced the risk of destruction of adaptogenic and biologically active components.

In future studies, to determine the mechanisms of formation and preservation of the functional properties of fermented plant beverages stabilized with microemulsions, it will be important to evaluate the survival of various probiotic microbial cultures in the plant substrate of the final product.

**Author Contributions:** Conceptualization, S.M.; Methodology, S.M. and O.Z.; Investigation, S.M. and O.Z.; Data curation and analysis, S.M., I.P. and O.Z.; Supervision, S.M.; Writing—original draft, S.M. and O.Z.; Writing—article and editing, S.M., O.Z. and I.P. All authors have read and agreed to the published version of the manuscript.

**Funding:** This research was partially supported by the grant RSF 22-26-00079.

**Institutional Review Board Statement:** Not applicable.

**Informed Consent Statement:** Not applicable.

**Data Availability Statement:** Data will be made available on request to the corresponding author.

**Acknowledgments:** We would like to thank the managers of Nanotechnology Research & Education Center of South Ural State University for their technical support during this work.

**Conflicts of Interest:** The authors declare no conflict of interest. The funders had no role in the design of the study; in the collection, analyses, or interpretation of data; in the writing of the manuscript, or in the decision to publish the results.

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
