# Peer review of "Fermented Plant Beverages Stabilized with Microemulsion: Confirmation of Probiotic Properties and Antioxidant Activity"

_fermentation, doi:10.3390/fermentation8120723_

Round 1

Reviewer 1 Report

The results of the research have proved the possibility of obtaining fermented plant beverages that do not contain allergenic factors, with identified probiotic and antioxidant properties. The data obtained allowed to establish a positive effect of stabilizing micro-emulsion loaded with biologically active components on the development of probiotic microorganism cultures in the system of fermented plant products and the formation of their antioxidant activity. Stabilization of emulsions with microcellulose additionally loaded with curcumin due to their prebiotic properties stimulates the development of beneficial microflora in beverages. In our study this was experimentally proved by analyzing the viability of probiotic microorganisms

1.     Why do you want to use microemulsion for fermentation?

2.     The authors should check the wrong spellings or the grammars through the whole manuscript, such as: line 252: It is was……

3.     I suggest the abbreviations should be repeated in the title of table, such as: ME, MCC, Twin…… It will help the readers for understanding easier.

4.     Is the Twin in the manuscript refer to twin-80? It can be confused to the readers by Twin.

5.     Is the Twin-80 will bring some disadvantages to us for our health in the food?

6.     Can you explain some mechanisms for the advantages in the microemulsion during fermentation?

Author Response

Dear Sir/Madam,

On behalf of my co-authors, I would like to acknowledge the valuable comments and suggestions proposed to enrich the content of this manuscript.

Every comment has been carefully evaluated and corrected. We hope that the revised article will meet your expectations.

All corrections made to the manuscript were marked using the "Change Tracking" function.

Best regards

S. Merenkova

Reviewer 2 Report

The research work is aimed to show the probiotic properties and antioxidant activity of fermented plant beverages stabilized with microemulsion. The introduction is too long. It should be more focused on the purpose of the work.   In the materials and methods section authors should:   1.       Enter the name of the expert who identified the seeds and where a sample of the plant species was preserved.   2.       On p3, line 182, the authors should enter where the microcellulose and TWEEN were purchased.   3.       Regarding the DPPH and TFC methods, the authors should report the reference methods or validate the ones they used   Finally, antioxidant activity can be monitored by a wide variety of assays with different mechanisms, including HAT (H atom transfer), ET (e- transfer), and mixed-mode (ET / HAT) assays, generally without distinct boundaries between them. Authors should report at least one other method of measuring antioxidant activity . The work is exciting, and I believe it should be published after major revisions.

Author Response

Dear Sir/Madam,

We appreciate the reviewers for putting a lot of time into reading and commenting on our manuscript. We are very much thankful for the critical comments and constructive suggestions, which helped to improve the manuscript quality.

We have revised our manuscript based on the reviewer’s comments. All corrections made to the manuscript were marked using the "Change Tracking" function.

Best regards

Svetlana Merenkova

Reviewer 3 Report

Honoured Authors,

please follow all my comments and suggestions in the enclosed Manuscript. 

Best wishes!

Author Response

Dear Sir/Madam,

We appreciate the reviewers for putting a lot of time into reading, editing, and commenting on our manuscript.

On behalf of my co-authors, I would like to acknowledge the valuable comments and suggestions proposed to enrich the content of this manuscript.

Every comment has been carefully evaluated and corrected. We hope that the revised article will meet your expectations. All corrections made to the manuscript were marked using the "Change Tracking" function.

Best regards

Svetlana Merenkova

Round 2

Reviewer 1 Report

accept.

Reviewer 2 Report

I believe that a single method for determining antioxidant activity is insufficient to determine the potential of the sample. Authors are advised to determine activity by ABTS, FRAP, or otherwise before publication.